# Multiple Targets for Oxysterols in Their Regulation of the Immune System

**DOI:** 10.3390/cells10082078

**Published:** 2021-08-13

**Authors:** Lisa Reinmuth, Cheng-Chih Hsiao, Jörg Hamann, Mette Rosenkilde, John Mackrill

**Affiliations:** 1Laboratory for Molecular Pharmacology, Department of Biomedical Sciences, University of Copenhagen, Blegdamsvej 3B, 2200 Copenhagen, Denmark; lisa.reinmuth@sund.ku.dk; 2Department of Experimental Immunology, Amsterdam Institute for Infection and Immunity, Amsterdam University Medical Centers, Meibergdreef 9, 1105AZ Amsterdam, The Netherlands; c.hsiao@amsterdamumc.nl (C.-C.H.); j.hamann@amsterdamumc.nl (J.H.); 3Neuroimmunology Research Group, The Netherlands Institute for Neuroscience, 1105BA Amsterdam, The Netherlands; 4Department of Physiology, School of Medicine, BioSciences Institute, University College Cork, College Road, Cork T12 YT20, Ireland

**Keywords:** oxysterols, ion channels, immune response, pharmacology, structure-function, inflammation, autoimmunity, infectious diseases

## Abstract

Oxysterols, or cholesterol oxidation products, are naturally occurring lipids which regulate the physiology of cells, including those of the immune system. In contrast to effects that are mediated through nuclear receptors or by epigenetic mechanism, which take tens of minutes to occur, changes in the activities of cell-surface receptors caused by oxysterols can be extremely rapid, often taking place within subsecond timescales. Such cell-surface receptor effects of oxysterols allow for the regulation of fast cellular processes, such as motility, secretion and endocytosis. These cellular processes play critical roles in both the innate and adaptive immune systems. This review will survey the two broad classes of cell-surface receptors for oxysterols (G-protein coupled receptors (GPCRs) and ion channels), the mechanisms by which cholesterol oxidation products act on them, and their presence and functions in the different cell types of the immune system. Overall, this review will highlight the potential of oxysterols, synthetic derivatives and their receptors for physiological and therapeutic modulation of the immune system.

## 1. Introduction

Oxysterols are cholesterol oxidation products, which can be absorbed from the diet, or generated by auto-oxidation or by enzymatic mechanisms. Oxysterols result from oxidation of cholesterol on the sterol rings, the side chain, or both [1]. This generates a diverse range of oxysterol congeners that have distinct biophysical properties. In addition to being reaction intermediates in the synthesis of bile-acids and steroid hormones, many oxysterols are biologically active signalling molecules, regulating diverse cellular processes. Consequently, the dysregulation of oxysterol production and action is associated with a range of diseases, including cancers [2], atherosclerosis [3], age-related macular degeneration, neurodegeneration and osteoporosis [1]. The focus of the current review are the roles of oxysterols in immune system physiology and pathology [4,5,6].

Oxysterols exert their biological effects in a variety of ways, potentially including mechanisms that are yet to be elucidated. Most oxysterols have direct biophysical effects on lipid bilayers, including the modification of cholesterol organization and associated membrane subdomains [7]. Several side-chain oxysterols, including 25-hydroxycholesterol (25-HC, synthesized by cholesterol 25-hydroxylase) and 27-hydroxycholesterol (27-HC, generated by cytochrome p450 oxidase 27A1, *CYP27A1*), play roles in the innate immune system, blocking the infection and replication of viruses and bacteria. For example, 25-HC hinders the replication of SARS-CoV-2 by inhibiting membrane fusion [8]. It also alters the mechanical properties of lipid membranes, leading to local variations in stiffness, which can influence the entry and egress of pathogens [9]. Ring-modified oxysterols, such as 7-ketocholesterol (7-KC), also inhibit viral replication and infection, but exert robust cytotoxic effects on host cells [10]. Oxysterols can indirectly modify membrane properties: 25HC activates acyl-CoA:cholesterol acyltransferase (ACAT) in macrophages, leading to the rapid internalization of an accessible pool of cholesterol, thereby blocking infection by *Shigella flexneri* or *Listeria monocytogenes* [11].

Since oxysterols are hydrophobic, they readily diffuse into cells to interact with intracellular receptors. Oxysterol-binding proteins (OSBPs) and OSBP-related proteins (ORPs) are a family of cytosolic receptors for oxysterols. In humans, the OSBP/ORP family is encoded by 12 distinct genes, with all members possessing an OSBP-related ligand-binding domain and the majority having a pleckstrin-homology domain near the N-terminus, that allows for interaction with phosphatidtidylinositol phosphate lipids. It is thought that all OSBPs/ORPs participate in the transfer of sterols and other small molecules at sites of membrane contact between different organelles [12], but additional roles have also been identified, some of which are specific to particular members [13,14]. Several of these roles were first identified in cells derived from the immune system. Inositol 1,4,5-trisphosphate (IP_3_) is generated upon the activation of G protein-coupled receptors (GPCRs) or receptor tyrosine kinases, through the action of phospholipase C (PLC) isozymes on the lipid phosphatidylinositol-4,5-bisphosphate (PIP_2_). IP_3_ is a second messenger that releases Ca^2+^ from intracellular stores, such as the endoplasmic reticulum (ER), via IP_3_ receptor (IP_3_R) channels. Oxysterols exert congener- and cell-type-selective effects on cytoplasmic Ca^2+^ levels [15]. In T cell acute lymphoblastic leukemia cells, ORP4L assembles the receptor CD3ε, its transducers Gα_q/11_ and the effector PLC-β3 into a signalling complex, generating IP_3_, which triggers Ca^2+^ release via IP_3_Rs. This increases mitochondrial matrix Ca^2+^, thereby activating oxidative respiration [16]. In the Jurkat T-lymphoma cell line, ORP4L interacts directly with IP_3_Rs, enhancing IP_3_-induced Ca^2+^ release [17]. In mouse bone marrow-derived mast cells, ORP9S is phosphorylated by a type 2 phosphoinositide-dependent protein kinase (PKD-2, most likely a protein kinase C-β (PKC-β)) and inhibits phosphorylation of Akt/protein kinase B at S473 [18]. OSBPs/ORPs play roles in viral infection and replication. For example, the drug itraconazole inhibits the replication of a diverse range of viruses, including hepatitis C and poliovirus, via the antagonism of the lipid transfer activities of OSBP and of ORP4 [19].

A key class of intracellular receptors for oxysterols exert their cellular effects through changes in the transcription of target genes [20]. Archetypal nuclear receptors for oxysterols are the liver X receptors (LXRs), which exist as heterodimers with retinoid X receptors (RXRs). Of the two human LXRs, LXRα (nuclear receptor subfamily 1, group H, member 3 (NR1H3)) is abundant in macrophages, liver, intestine, adipose tissue, lung, kidney and adrenal gland, whereas LXRβ (NR1H2) is ubiquitous [21]. Binding of certain side-chain oxysterols (22(R)-HC, 24(S)-HC, 25-HC, or 27-HC) elicits conformational changes that promote dissociation of LXR-RXR complexes from transcriptional co-repressors, and stimulates LXR-RXR interaction with the sterol response elements (SREs) of target genes. These mechanisms increase the transcription of genes involved in cholesterol clearance (ATP-binding cassette A1 and G1, *CYP7A1*, apolipoprotein E) and those which regulate multiple components of the immune system [4].

There are three widely expressed members of the SRE-binding protein family (SREBP1a, SREBP1b and SREBP2), which are transcription factors tethered to the ER via a transmembrane-spanning domain. In the ER, SREBPs interact with SREBP cleavage-activating proteins (SCAPs). When the levels of cholesterol and oxysterols are low, SCAP transports SREBPs to the Golgi apparatus where it is cleaved by two resident proteases, releasing an N-terminal domain. This domain translocates to the nucleus, where it binds to SREs in the promoters of target genes [22]. These target genes include those involved in fatty acid metabolism, cholesterol transport and immune responses [6]. Cholesterol, or certain oxysterols (most notably 25-HC), bind to the ER-resident proteins insulin-induced gene-1 (INSIG-1) or INSIG-2, promoting interaction with SCAP [23], preventing its translocation to the Golgi and activation of SREBP. Consequently, cholesterol and some oxysterols are negative modulators of SREBP-dependent transcription.

The RXR-related orphan receptor (ROR) family members α and γ can bind to certain oxysterols, particularly 7-position substituted congeners, with high affinity. This activates the transcription of target genes, including those involved in immunity. Other oxysterols, such as 24S-hydroxycholesterol (24S-HC) and 24S,25-epoxycholesterol, suppress the activities of these transcription factors [24].

Several oxysterols are selective estrogen receptor modulators. For example, 27-HC binds to both the α- and β-estrogen receptors, inducing transcription of their target genes [25]. In contrast, 27-HC antagonises the protective effects of estrogen against atherosclerosis in a diet-induced mouse model of hypercholesterolaemia [26].

Certain oxysterols influence the transcription of target genes via epigenetic mechanisms. During investigations of the effects of high extracellular glucose on the regulation of genes involved in non-alcoholic fatty liver disease, it was discovered that elevated 25-HC promoted methylation of CpG islands, acting as a direct, endogenous agonist of DNA methyltransferase 1 (*DNMT1*) [27]. Genes whose transcription is regulated in this manner include voltage-gated Ca^2+^ channel subunits, Ca^2+^-calmodulin kinases, those involved in lipid metabolism, inflammation and MAPK signalling. Within the immune system, cholesterol 25-hydroxylase, the enzyme that produces 25-HC, is particularly abundant in macrophages, and this oxysterol suppresses IgA synthesis [28]. In addition, 25-HC can be sulfated at its 3-position by sulfotransferase 2B1b to generate 25-HC3S, which can be further sulfated by sulfotransferase 2A1 [29]. In contrast to 25-HC, 25-HC3S is a potent antagonist of DMNT-1, -3a and -3b, and opposes its actions on target genes [27,30,31]. In a lipopolysaccharide-induced model of acute lung injury, 25-OHC concentrations are elevated, potentially promoting inflammation and the recruitment of leukocytes [32]. Since 25-HC3S has beneficial effects in models of non-alcoholic fatty liver disease, organ damage and inflammation, and has been evaluated favourably in clinical trials, further elucidation of the roles of this oxysterol metabolite in the regulation of immune cells will be of great value [31].

In addition to the direct effects of oxysterols on nuclear receptors, there is also considerable cross talk between signalling pathways modulated by cholesterol oxidation products and those regulated by other bioactive sterols. Some of these sterols are important modulators of the immune system. For example, active vitamin D (1,25-dihydroxycholesterol) inhibits the expression of *CYP7A* (cholesterol 7 hydroxlase α) by promoting the formation of a complex between the vitamin D receptor and LXRα [33]. The enzyme 11β-hydroxysteroid dehydrogenase type 1 (11β-HSD1) can convert 7-KC to 7β-hydroxycholesterol (7β-HC), and also interconverts active cortisol (a glucocorticoid hormone) into inactive cortisone. In adipocytes, co-incubation with 7-KC repressed, and 7β-HC enhanced, the biological effects of cortisol through a mechanism that involved changing the direction of substrate flux through the 11β-HSD1 and 11β-HSD2 enzyme pair [34].

Despite the potential for therapeutic exploitation of nuclear receptor-mediated oxysterol signalling in the regulation of immune cell biology, this does have two key limitations. Firstly, transcriptional mechanisms are slow, with a lag of tens of minutes to hours between the initial stimulus (change in the levels of a particular oxysterol) and the cellular response. This may be unsuitable for the control of cellular processes that can occur over much shorter timescales, such as motility, endocytosis and secretion. Secondly, in terms of the development of oxysterol-based immunomodulatory drugs, LXR agonists act on the liver to increase the expression of SREBP1c, which promotes the clinically unfavourable condition of hypertriglyceridemia [6,35,36]. Targeting the main focus of the current review, cell-surface receptors for oxysterols might provide effective solutions to these limitations. These targets can be divided into two classes: G protein-coupled receptors (GPRs) and ion channels. Evidence for the presence of these proteins in different immune cells was derived from two key sources: analyses of four independent transcriptomic databases from human immune cells (Appendix A) [37,38,39] and a review of the scientific literature (Appendix A).

## 2. G Protein-Coupled Receptors

G protein-coupled receptors (GPCRs) are predominantly located in the cell-surface membrane. They have a characteristic 7-transmembrane (7TM) topology and regulate the activities of second-messenger-producing enzymes. Figure 1 summarizes some of the properties of the three types of GPCR that are present in the mammalian immune system and which interact with oxysterols. Although there are four types of GPCR that are known to be modulated by oxysterols, one of these, GPR17, is undetectable in the immune cell types that we investigated, see Appendix A. Figure 2 provides an overview of the distribution of these receptors among cells of the immune system.

### 2.1. Epstein–Barr Virus-Induced Receptor 2, or GPR183

#### 2.1.1. Cellular Expression Pattern and Overall Biological Function

Epstein–Barr virus-induced receptor 2 (EBI2), or GPR183, is mainly expressed by immune cells and related tissues. Secondary lymph organs have an especially abundant expression, but the lungs and the gastrointestinal tract also show comparably high expression levels [40].

The best-established role for the receptor lies in the positioning of cells in secondary lymph organs, and especially the facilitation of T cell-dependent antibody responses. Through the interplay and differential expression of enzymes in the secondary lymphoid organs, a gradient of 7α,25-OHC is created, with the highest concentration at the B/T-cell border and the outer follicular regions [41]. B cells undergo the sequential up and down regulation of GPR183, in parallel with the expression of several other chemotactic GPCRs. Activation of B cells by B-cell receptor (BCR) binding leads to an upregulation of GPR183. This, in turn, results in the lymphocyte alignment on the T-cell border, a high ligand concentration region, by integrated signalling via both GPR183 and C-C chemokine receptor 7 (CCR7) [42,43]. Here, they can receive secondary differentiation signals from T-helper (Th) cells. If they are T cell-stimulated, GPR183 expression is downregulated, which allows the B cells to move to the germinal centres in the B-cell zone for antibody refinement or escape to the circulation. Expression in the germinal centre is generally lowered, but the differential expression of GPR183 still correlates with localization in the light or dark zone [43,44]. Memory B cells, on the other hand, stay on the fringes of the follicular area, due to high GPR183, normal CCR7 levels and downregulated CCR5 expression. By a similar mechanism, corresponding specialized TH cells are localized to the same region [45].

T cells also receive cues from oxysterols via GPR183. In their development process, GPR183 influences central tolerance, especially in CD4 single positive (SP) cells. It regulates CD4^+^ effector and regulator cell generation by increasing encounters with antigen-presenting cells, such as dendritic cells (DCs), and thus increases the number of encounters with different, rare self-antigens [46]. GPR183 remains highly expressed on CD4^+^ T cells compared to CD8^+^ T cells and also enhances CD4–DC encounters in secondary lymph organs. Thus, it enables the early differentiation of CD4^+^ helper cells [47], which in turn facilitates more efficient CD8^+^ T- and B-cell responses. In exemplum, GPR183 stimulates T follicular helper cell (TfH) differentiation and then induces secondary migration to the follicles [48].

Bringing it back full circle, DCs also express GPR183, and react to oxysterol cues by migration. This has mainly been studied in context of correct positioning in the spleen, in the bridging channels unto activation [49]. In activated DCs GPR183 is also responsible for localization at the outer T zone, where they determine TH cell differentiation to TfH cells [50]. GPR183 expression has also been found relevant in plasmacytoid (pDC) and myeloid (mDC) subsets, where it downregulates interferon responses upon Toll-like receptor (TLR) binding, as well as migration towards the spleen [51]. These regulatory effects are also supported by an observed interplay between tuberculosis infection and type two diabetes, where reduced GPR183 expression correlated with increased interferon levels and infection severity [21].

The expression of GPR183 in eosinophils, together with increased oxysterol production in inflamed airways, could be responsible for recruitment to the tissue [52]. Another specialized immune compartment is the gastrointestinal tract, where lymphoid tissue inducer (LTi)-like group 3 innate lymphoid cells (ILC3) have been found to be organized in a GPR183 dependent manner and this affects inflammatory processes in the bowel [53,54].

GPR183 is also expressed on astrocytes and microglia in the brain, where it may thus effect neuroinflammatory diseases [55,56,57,58]. In the context of neuropathic pain, GPR183-expressing astrocytes and microglia accumulate in the spinal cord and allodynic effects can be attenuated by the inhibition of this receptor in vivo [59].

Related to this, several studies have looked at the effects of the GPR183–oxysterol axis in multiple sclerosis. GPR183 dependent T-lymphocyte trafficking seems to be of significant relevance here. In particular, T_H_17 and memory lymphocytes were attracted by high 7α,25-OHC concentrations, which can be established by activated microglia in inflammatory lesions [60,61,62].

Unrelated to the immune system, GPR183 is also expressed on monocyte/osteoclasts precursors, and thus heightened 7α,25-OHC secretion by osteoblasts, and in an autocrine manner by osteoclast precursors themselves, provides signals for bone resorption [63]. This might also be of relevance in the control of bone mineral density during systemic inflammatory or high cholesterol settings.

#### 2.1.2. The Chemistry and Production of Endogenous Oxysterols That Bind GPR183

7α,25-OHC has been determined to be the endogenous ligand with the highest affinity and strongest activity for GPR183, though other oxysterols like 7α,27-OHC or 7β,25 OHC also activate this receptor, albeit with lower potency and efficacy [64]. 7α,25-OHC is synthesized from cholesterol in two steps. The first additional hydroxyl group is added in position 25 facilitated by cholesterol-25 hydroxylase (CH25H); the second is catalysed by *CYP7B1* [65]. Alternatively, 7α,25-OHC can be produced by a combination of *CYP7A1* and *CYP3A4* [66,67]. However, this is insufficient to launch normal T cell-dependent antibody responses [64].

The gradient in the lymph nodes stems mainly from stromal cells. Here, the cell-specific knockout of these key enzymes disrupted B-cell positioning. Several hematopoietic cell types, for example, follicular DCs, have also been shown to express Ch25H and Cyp7b1, but ligand expression in the lymph tissue was not affected by the lack of these enzymes in bone marrow-derived cells [41]. This becomes relevant in normal tissue immune responses, as the key enzymes can be differentially regulated. Innate immune cells, such as DCs and macrophages, increase Ch25H expression via multiple immune pathways [68]. Lipopolysaccharide (LPS) challenge of mouse macrophages, mimicking bacterial infection, resulted in heightened Ch25H expression and, correspondingly, increased the plasma concentration of 25-HC in human test subjects [69]. Human primary monocytes express both receptor and ligand producing enzymes, with a marked upregulation upon LPS stimulation [70]. The lungs are suggested to be a tissue where this holds central significance, as *CH25H* is especially highly expressed and LPS challenge in KO mice results in prolonged neutrophilia [71]. Epithelial cells in the airways also upregulate these oxysterol-producing enzymes in chronic obstructive pulmonary disease conditions. Here, the higher concentration was shown to attract B cells via GPR183 (and stimulate local lymphoid tissue formation) [72]. An older study also found a correlation between increased 25-HC secretion and decreased IgA class switching, with the same effect in *CYP7B1* KO mice [28]. Under normal physiological conditions, 25-HC and 27-HC concentrations are increased in Cyp7b1 KO mice but, otherwise, the phenotype remains mostly normal. Enzyme function in bile acid synthesis seems to be substitutable [73].

*CYP7B1* expression is regulated by nuclear factor kappa B (NF-κB) and the Janus kinase signalling pathway, tying it to inflammatory and metabolic stress contexts. In rheumatoid arthritis, for example, upregulation of tumour necrosis factor alpha (TNFα), interleukin 1 (IL-1) and IL-17 induces heightened *CYP7B1* expression by fibroblastic cells [74]. During acute lung inflammation Cyp7b1 is also upregulated [75]. In insulin resistance states, on the other hand, *CYP7B1* expression is low, while GPR183 and *CH25H* are upregulated, leading to accumulation of intermediate oxysterols in non-alcoholic fatty liver disease [76,77]. However, a direct correlation between disease progression and GPR183 signalling was not found. Breast cancer cells downregulate *CYP7B1* expression by methylation, and promote CYP27A1 expression by specifically differentiating and recruiting M2 macrophages, leading to a feedback loop increasing 27-HC concentrations. This promotes tumour growth and metastasis [78].

Hydroxysterol dehydrogenase-3 B7 (*HSD3B7*) is an enzyme relevant in the degradation of 7α,25-OHC and a number of other oxysterols to form bile acids. *HSD3B7* is expressed in secondary lymph organ stromal cells and fDC and is necessary to create a balanced gradient [41]. It has also been found in the intestines, especially in stromal cells surrounding the lymphoid structures [53]. Interestingly, the cue for fDC localization itself is given by 7α,27-OHC, which is produced by another enzyme, Cyp27a1, also expressed in stromal cells, but also degraded by HSD3B7 [50]. This might be another mechanism to achieve differential signalling.

Overall, these possible relation certainly shows there is room to explore unravelling the signalling pathways governing the immune system and, in particular, the antibody production capabilities.

#### 2.1.3. GPR183 Receptor Structure

To date, there is no resolved receptor structure. However, several attempts at homology modelling have been made, accompanied by extensive mutagenesis studies. Developments in this structure guided approach, with the goal of advancement towards targeted drug discovery, were mainly made by two groups: those of Changlu Liu and of Mette Rosenkilde. Confirmation of a specific binding mode of either the endogenous ligands or existing small molecules remains elusive.

Mutagenesis efforts concentrated on residues in the orthosteric, major and minor binding pocket area, where several amino acids showed effects on signal transduction to Gαi or β-arrestin. Since GPR183 had been described as an orphan receptor until around 2010, several mutagenesis studies were performed, where either the inhibition of supposed constitutive activity occurred, or where binding of the first synthetic compound were analyzed [79,80]. By these means, an initial set of functionally important residues was identified. The strongest effects of mutants were seen in R87, increasing activity, and F257 of the CFxP-motif, decreasing it. Several partially conserved chemokine receptor (CXCR) motifs were also detected, in conformity with GPR183 now generally believed to be closely related to that receptor family. Both groups proposed binding poses on homology models based on the phylogenically closest known resolved structure of CXCR4 (PDB:3ODU) or a CXCR4 chimera, but those differ [81]. In the model from Zhang et al., the key interacting residues were proposed to be R87, N114 and E183 [81], of which N114 was later proven to belong to an allosteric metal ion site. However, the groups did agree on the relevance of four residues: R87, Y112, Y116, Y260, which were determined as the key binding residues in the model from Benned-Jensen et al. [82]. A third homology model based on CXCR4 exists, made to compare the orthosteric sites of the three class A GPCRs characterized to be oxysterol binding [83]. The one residue not appearing in their binding site prognosis is Y116, since it is rotated in a different direction in the template structure. The modelling of GPR183 on more recent crystal structures includes the PAR1 receptor, where possibilities for biased agonism were in focus. In this process, the receptor has been shown to contain the conserved allosteric sodium binding site found in several class A receptors, with special importance of the main residues D77, N114 and D304 [84].

Further analysis to confirm the specific binding residues is in the making though, to improve identification opportunities for pharmacologically relevant new drug candidates. These days, there are several endogenous oxysterols and small molecule modulators, but a definite site for neither is known. Recent discoveries include a number of allosteric sites in class A GPCRs formerly unconsidered [85,86]. Of special interest are binding sites outside of the helices, in the lipid bilayer, since oxysterols, with their highly lipophilic character, would have easy access here. This is also supported by cholesterol molecules, which favour specific sites in resolved structures or even enter receptors from here [87].

### 2.2. CXC Chemokine Receptor 2 (CXCR2)

#### 2.2.1. Cellular Expression Pattern and Overall Biological Function

CXCR2 is a chemokine receptor. It is phylogenetically related to CXCR1, with which it shares high sequence similarity and has some overlapping ligands and mechanisms. CXCR2 is highly expressed by innate immune cells and endothelia. One of the main functions is neutrophil recruitment and activation, leading to the creation of a pro-inflammatory environment to promote pathogen clearance. In cancer, neutrophils recruited by this mechanism can also further promote tumour growth and metastasis by the induction of neoangiogenesis and inhibition of cytotoxic T cells [88,89]. Opposing effects have also been demonstrated. Angiogenic effects might also be attributed to CXCR2 expression on endothelial cells, which also leads to chemotaxis in these cells [90].

Several other diseases have been associated with CXCR2 signalling, mostly related to neutrophil activity. Among these are neurodegenerative diseases, such as multiple sclerosis, where increased ROS production leads to sustained neuronal damage [91] and neutrophil activation facilitates blood–brain barrier breakdown [92]. Several other inflammatory diseases are influenced by CXCR2 dysregulated neutrophils. Pancreatitis can be attenuated by the inhibition of CXCR2 signalling on neutrophils, where the main effects are attributed to decreased neutrophil chemotaxis-reducing activation of monocytes and, thus, cell death in the pancreatic tissue [93]. The same holds true in ulcerative colitis, where attenuated chemotaxis leads to a decrease in inflammatory cytokine expression [94,95].

CXCR2 is also expressed in hematopoietic stem cells, where signalling plays a role in homeostasis [96]. It also acts as a mobilization signal for neutrophil release from the bone marrow [97]. Furthermore, the fate of differentiating cells of the myeloid line is dependent on the receptor and a bias towards suppressor cells exists in ligand rich tumour settings [98].

#### 2.2.2. The Chemistry and Production of Endogenous CXCR2 Ligands

CXCR2 has several endogenous ligands. The receptor is sensitive to all ELR^+^ CXC chemokines to varying degrees [99]. Different sets of these are expressed in different settings, but CXCR2 is recognized as their main receptor. CXCL1 and 2, for example, provide the cue for mobilization of neutrophils from the bone marrow [97] and are contributing factors in CNS autoimmune diseases due to increased immune cell infiltration upon TH17 induced chemokine upregulation [92]. CXCL8, on the other hand, plays a significant role in recruitment to sites of infection and vascular escape [100]. Here, it has been shown that concentration, steepness of the gradient, as well as ligand conformation and complexation, play differential roles in signalling outcomes.

Interestingly, in non-inflammatory, non-disease settings a key regulator of endothelial ELR^+^ CXCL expression seems to be low shear stress as seen in capillaries Shaik et al., [101]. This is regulated via p38, NF-κB signalling pathway, which relates well to increased expression in inflammatory settings.

Oxysterols are another wide group of endogenous ligands for CXCR2. A broad range of single hydroxylated cholesterol derivates proved to be able to induce migration in neutrophils [102,103]. The focus in this study was on 22-HC, due to its higher concentration in tumours. The group reported higher neutrophil recruitment of CXCR2-expressing cells, accompanied by the described pro-tumour effects. They did not, however, see synergistic effects between oxysterols and endogenous CXCLs. The effects seem to be independent from each other, with oxysterols sufficient to induce chemotaxis. A later study also elucidated the effects of 27-HC [89], the second highly abundant oxysterol in cancer. In comparison, they observed increased metastasis with 27-HC treatment. This effect was facilitated by γ/δ T cells, next to the intrinsically important neutrophils, through a vicious cycle between these populations.

#### 2.2.3. CXCR2 Structure

Since antagonism of CXCR2 has been suggested as a therapeutic tool in a wide spectrum of issues for several decades, a lot of effort was put into finding, refining and proving the concepts behind a wide panel of compound classes. Methods range from early chimeric proteins and site-directed mutagenesis to partial structure resolution. Finally, in 2019, a full receptor structure, including an endogenous ligand and G protein and a version with a small molecule, were resolved (PDB:6LFL/M/O) [104].

Mutagenesis studies revealed several distinct areas that are responsible for the binding of different ligands. Endogenous CXCLs are thought to bind on the extracellular side, with an emphasis on N-terminal involvement for CXCL1 and probably other ELR^+^ CXC chemokines not inducing signalling on CXCR1, and increased importance of ECL2 for binding of CXCL8 [105].

Known CXCR2 antagonists can be divided into several classes, which seem to, at least in part, bind distinct sites on the receptor. Two main sites have been described, one intracellular, possibly interfering with G protein binding, and one inside the transmembrane helical bundle, probably interfering with activation-related conformational changes [106,107,108]. This intrahelical pocket is located inside of TM 3, 5 and 6, with an entry point from the TM5/6 interface. Interestingly, the environment of the entry channel seems conductive for lipid binding, as seen in the CXCR4 crystal structure, which one might speculate to be a possible site for interaction with oxysterols too.

The intracellular C-terminus of CXCR2 carries a PDZ binding motif, which is relevant for signal transduction via Na^+^/H^+^ exchanger regulatory factor (NHERF1), which is especially relevant in neutrophil chemotaxis. Interference with this interaction has also been proven to modulate disease mechanisms [109,110].

The full length cryo-EM receptor structures resolved by Liu et al., 2020 [102] further elucidated the binding of CXCL8, G_i_ and an antagonist at the intracellular site, overlapping with the G-protein site. CXCL8 turned out to have a fairly shallow binding compared to other chemokine receptor/ligand complexes. In comparison with CXCR1, two major interaction sites are likely key to permissive ELR^+^ CXCL recognition, the PP-PC motif region on the N terminus and CSR2 that directly interacts with the ELR motif. Another interesting finding was a resolved cholesterol located on the TM 2, 3, 4 interface at the height of the inner membrane leaflet in the active state structure. This might indicate a different oxysterol binding site, stabilizing the receptor in the active conformation. There have been studies looking at possible effects of cholesterol in general chemokine receptor function [111], and some of the mechanisms might be correlated or overlapping.

### 2.3. Smoothened (SMO)

#### 2.3.1. Cellular Expression Pattern and Overall Biological Function

Smoothened (*SMO*) is a class F GPCR best known for its actions in development and cancer. It is an oncoprotein and part of the hedgehog (Hh) signalling pathway, where it is responsible for transmembrane signal transduction. This happens when Hh binding to its receptor Patched relieves inhibition of SMO. Several direct ligands and modulators for SMO have been known for a while [112,113]. These can activate or inhibit different steps in the activation cascade, before relocalisation to the basal cilia, or at the stage of final activation [114]. Signalling results in Gli family transcription factor cleavage, where pathway activation leads to activator fragments, while downregulation increases the repressor fragment availability. There are three different mammalian hedgehog proteins, and three Gli factors.

Hh signalling also acts as a regulator in the immune system, Figure 3, and has direct regulatory influences on the adaptive side. Lymphocyte development and fate can be dependent on the Hh pathway and thus on SMO, Figure 3. Differential Hh signalling has been shown to be important in various steps of T cell differentiation, as well as effector function. In the development sonic hedgehog (SHh)-dependent Gli activation is important for transition of DN1 to DN2 and survival of DN4 cells. Intermediate concentrations accelerate transition to DP cells, while very high or very low concentrations lead to arrest [115,116]. In contrast, DN3 survival and later proliferation of DP to SP effector cells increased upon downregulation of Hh pathway activation [117,118].

Shh levels also influence cell fate. Homeostasis and maturation of γ/δ T cells in adult mice are reduced upon inhibition of Hh signalling [119,120]. Low SHh increases the rate of differentiation to single positive cells, while skewing balance between CD4^+^ and CD8^+^ towards CD4^+^ T cells [120]. CD4^+^ cells are themselves skewed towards Th2 differentiation upon higher signalling via Gli2A [121]. Indian hedgehog (IHh), another Hh ligand, is needed for efficient CD8^+^ cytotoxicity [122]. Natural killer T (NKT) cells, another T-cell subpopulation, also possess an increased cytotoxic profile upon SHh stimulation, which they can produce themselves, thus opening the possibility of entering an autocrine loop [123].

Other cells of the adaptive immune system are also subject to Hh signalling. Early B-cell development is dependent on SHh, such that lower levels stimulate differentiation, while higher levels reduce B cell commitment and maturation. This is regulated via Gli3 repression of SHh expression in the stromal cells [124,125]. A similar mechanism, as has been shown for thymic epithelial cells [120] and used in skin inflammation experiments to check on the influence of Hh signalling on peripheral immune cells [126].

#### 2.3.2. Chemistry and Production of Endogenous SMO Ligands

There is a certain degree of speculation about the endogenous ligand of SMO. Neither the mechanism of how Patched inhibits SMO activity, nor which lipid or small molecule might be endogenously responsible for signal transduction, has been elucidated so far. Sterols have been proven to modulate SMO activation though, and their widespread natural occurrence qualifies them as possible endogenous ligands or synergistic allosteric modulators.

In vitro studies found that oxysterol 20(S)-HC is the most potent agonist, though 22(S)-HC, 7-keto-25-HC and 7-keto-27-HC and others activate the receptor [127]. All of these have the normal cholesterol hydroxyl group in position 3 in common. Several of these are able to synergize with each other, or with some of the other known small molecule ligands. Unmodified cholesterol also binds SMO, and can activate the receptor, as well as synergizing with Hh signals or other ligands. An important relation between SMO activation and cholesterol biosynthesis was established long ago [128], but the question of whether cholesterol itself is the endogenous ligand has been raised multiple times over the years and remains unanswered. Fairly recently, a study showed that covalent cholesterol modification takes place on residue D95 in human SMO (in the extracellular CRD binding site) [129]. It might be that either cholesterol esterification, binding or oxysterol availability are regulated by Patched, or that they provide a second signal, maybe ensuring capability of the cell to support the induced changes in transcription. One way in which oxysterols could be enriched for SMO activation has been described by Raleigh et al. [130]. They found that a number of oxysterols, which can synergistically activate SMO via two sites, are enriched in the cilia membrane, where final SMO activation usually takes place.

#### 2.3.3. SMO Receptor Structure

There are several receptor structures for SMO, published dating back to 2013, using methods ranging from NMR to conventional X-ray diffraction, to cryo-electron microscopy. A number of ligands have been resolved, bound to fragments, the full-length receptor, or receptor G protein complexes. Especially relevant for this review are the structures containing cholesterols and derivates.

A cysteine-rich domain (CRD) crystal structure from [131] (PDB:5KZV) shows a binding site for 20(S)-OHC. The authors highlighted the binding mode and allosteric transition induced by binding and discovered key residues that are important for both oxysterol and cholesterol binding, which both induce conformational changes and are able to activate the receptor for/with SHh stimulation. The group hypothesized that cholesterol is the endogenous ligand, due to its physiologically relevant concentration. Further studies found that cholesterol seems to be covalently bound in that site at some point. Residue D95, which had been mutated by the aforementioned group and abolished binding in their assays, showed up with a correspondingly increased molecular weight in MS/MS. However, the mechanism of this modification and its explicit regulatory function remain unclear [129].

Comparing different structures resolved in recent years, some major differences can be seen in the tilt of the CRD region. This might be due to different activation states [132,133]. Deshpande et al. [134] solved a full receptor structure, stabilized in the active state by nanobody binding, with cholesterol bound in a deep pocket inside the 7TM region (PDB:6O3C). This binding produces the activating changes for GPCRs, independent of Hh, especially the marked shift in TM6. The binding pocket opens up with the receptor in a more active-like state; for example, with agonists bound higher up in the receptor, or by stimulation from the CRD site. The known antagonists bind at a site that overlaps partly with this deep cholesterol site, thus inhibiting activation this way as well as stabilizing the inactive state.

A year later, Qi et al. [135] solved more structures of WT and SMO-Gi complexes with several oxysterols and an agonist, SAG, bound (PDB:6XBM). In this way they discovered more, interconnected binding sites, which seem to allow for transfer through the receptor, from the deep 7TM pocket outwards to the CRD.

### 2.4. Gamma-Amino Butyric Acid (GABA) Type B Receptors

GABA is the main inhibitory neurotransmitter in the vertebrate central nervous system. It exerts biological effects by interacting with ionotropic GABA_A_Rs (see Section 3.1.3), which contain an intrinsic chloride channel, and with metabotropic GABA_B_Rs, which are GPCRs. There is limited evidence supporting a modulatory role for oxysterols in GABA_B_R signalling. In brain slices from the rat lateral septum, the superfusion of 25-HC selectively reduced GABA_B_R- but not GABA_A_R-dependent inhibitory post-synaptic potentials [136]. Microglia are known to possess both GABA_B_R subtypes. The activation of these receptors inhibits microglial IL-6 and IL-12p40 secretion in response to LPS [137]. The GABA_B_R-selective antagonist, baclofen, is reported to inhibit DC cell activation and their priming of Th17^+^ T cells [138].

## 3. Ion Channels

Gating of ion channels permits rapid changes in intracellular ion concentrations in response to diverse stimuli, leading to alterations in membrane potential, the second messenger Ca^2+^, cell volume, cell-death, gene expression, secretion, endocytosis, or motility. The gating of a limited range of ion channels is known to be modulated by oxysterols, often in a congener- and channel subtype-selective channel fashion. In addition to cell-surface channel proteins, IP_3_R Ca^2+^ release channels are located predominantly on the ER and are also modulated by oxysterols. This includes the stimulation of the proteolytic degradation of IP_3_Rs [139], and the assembly of IP_3_R signalling complexes by [16], and direct interactions with, ORP4L [17]. The cell-surface ion channels that are modulated by oxysterols can be divided into three functional categories: those gated by ligands; those gated by changes in membrane potential (voltage-gated); and those gated by multiple stimuli (multi-modal gating). The presence and roles of these channels in the immune system are largely unexplored. Their levels of transcription in immune cells are summarized in Figure 4. These channels have considerable potential as targets for the development of new therapies to combat immune disorders, including autoimmune diseases [4,5].

### 3.1. Ligand-Gated Ion Channels

These are gated by the binding of a specific ligand to a site within the ion channel complex. Only a limited number of this type of channel have been reported to interact with oxysterols: the P2X7 purinoreceptor (P2X7R), the *N*-methyl-d-aspartate receptor (NMDAR) subtype 2B (encoded by *GRIN2B*) and the GABA_A_Rs.

#### 3.1.1. The P2X7 Purinoreceptor (P2X7R)

P2X7R is a member of the P2X family of purinoreceptors that bind ATP, opening a non-selective cation channel that is highly permeable to Ca^2+^. P2X7R monomers contain two transmembrane helices and a cytoplasmic C-terminus that is extended relative to other members of the P2X family. P2X7R is distinct from other members of its family in that it is formed only by homotrimeric complexes, as opposed to heterotrimeric ones, and in that it binds ATP with low affinity. When stimulated by low concentrations of ATP, the P2XR channel attains a conformational state that is permeable to inorganic cations. When stimulated with higher concentrations of ATP for sustained durations, the channel pore dilates, or recruits additional components, such as pannexin-1, so that it forms a pathway that is permeable to much larger organic cations [140]. This dilated pore conformation is thought to underlie the subtype-specific cytotoxicity resulting from P2X7 activation. The molecular structures of rat P2X7R in both its apo, closed state (PDB:6U9V) and ATP-bound, open state (PDB:6U9W) have been determined using cryo-electron microscopy, but these provide no evidence supporting a dilated pore conformation. However, these structures indicate a role for cysteine palmitoylation in maintaining the channel in a non-desensitizing state [141].

P2X7R channel gating is modulated by several types of sterol. The depletion of cholesterol from the plasma membranes of cells transfected with P2X7R increases permeation of organic cations, whereas cholesterol loading inhibits this mechanism. An N-terminal domain and multiple cholesterol recognition amino acid consensus motifs near the C-terminus are key determinants of this cholesterol sensitivity [142]. Vitamin D (1,25-dihydroxycholesterol) rapidly increases in Ca^2+^ in peripheral blood mononuclear cells via a mechanism dependent on P2X7R [143], but it is not known if this effect is mediated by binding to the same sites as cholesterol. There is also evidence that oxysterols modify the gating of P2X7R channels. In a model of human skin, 25-HC promotes both caspase-dependent apoptosis and pyroptosis via a P2X7R-dependent mechanism [144]. In human retinal pigment epithelial cells, both 7KC and 25-HC activate P2X7R, with the latter oxysterol requiring pannexin-1 for this effect [145].

P2X7R is present in many cells of the immune system, with the highest levels in monocytes, followed by NK cells and CD4^+^ memory T_reg_ cells; Figure 4 and Appendix A. The roles of oxysterols in modifying P2X7R responses in immune cells are yet to be elucidated. This receptor plays a key role, as in the recognition of extracellular ATP, an archetypal damage associated molecular pattern (DAMP) molecule that is released from injured cells [146]. In monocytes, the acute stimulation of P2X7R enhances the secretion of IL-1β [147], whereas sustained activation promotes apoptotic cell death. Extracellular ATP promotes the activation of the NLRP3-inflammosome in monocytes and this process is impaired during sepsis [148].

Substantial experimental evidence supports the roles of P2X7R in T-lymphocyte biology [149]. P2X7R antagonists inhibit the TCR-dependent activation of mouse T cells, as assessed by their effect on IL-2 production [150]. Through P2X7R, ATP inhibits both the differentiation of naïve CD4^+^ cells into regulatory CD4^+^ CD25^+^ T cells (T_reg_) and the immunosuppressive capabilities of this subtype [151]. The activation of P2X7R drives the decision for differentiation to an αβ rather than a γδ TCR phenotype [152]. Human CD4^+^CD45RO^+^ memory T cells indirectly inhibit NLRP3 inflammasome activation in ATP-stimulated monocytes, via downregulation of P2X7R [153]. In response to *Plasmodium chabaudi* malaria infection, P2X7R stimulates Th-cell differentiation and decreases the Tfh subset [154]. Extracellular ATP promotes metabolic fitness in long-lived CD8^+^ memory T cells, through a mechanism involving P2X7R and AMP-activated protein kinase [155]. P2X7R stimulates non-classical secretion in numerous cell types, including T lymphocytes [156]. It also impairs the MHC-I-dependent presentation of oligopeptides by antigen presenting cells, thereby suppressing the activation of CD8^+^ T cells [157]. ATP drives apoptotic death in thymocytes via the activation of P2X7R. This receptor also underlies inflammation in a model of colitis, by promoting apoptosis in T_reg_ cells [158].

#### 3.1.2. Ionotropic Glutamate Receptors

Glutamate is the most abundant excitatory neurotransmitter in the mammalian central nervous system. Receptors for glutamate are divided between those that are ionotropic, containing an intrinsic ion channel, and metabotropic GPCRs. Ionotropic glutamate receptors are named after their definitive pharmacological ligands: *N*-methyl-d-aspartate (NMDA), γ-amino butyric acid (GABA) and α-amino-3-hydroxy-5-methyl-4-isoxazolepropionic acid (AMPA). Endogenous agonists of NMDAR channel complexes are glutamate, d-serine and/or glycine, depending on their subunit composition. They are heterotetrameric channel complexes, usually consisting of two obligatory GluN1 subunits (encoded by *GRIN1*) and two GluN2A, GluN2B, GluN2C, or GluN2D subunits (*GRIN2A, GRIN2B, GRIN2C,* or *GRIN2D*). Two alternative subunits, GluN3A and GluN3B (*GRIN3A, GRIN3B*), can combine with GluN1 to form a complex that has reduced current, which is gated by glycine and not by glutamate [159,160,161]. The opening of NMDAR channels usually results in the influx of Ca^2+^ and Na^+^, and an efflux of K^+^. In vivo, the permeation pathway can be blocked by Mg^2+^, with this inhibition being relieved by depolarization. NMDARs are predominantly present in the central nervous system, where they act as post-synaptic glutamate receptors. They have also been identified in multiple peripheral, non-neuronal cell-types [162,163]. The presence of NMDARs in cell types that circulate in the blood is potentially problematic, since the concentrations of amino acid agonists in the extracellular fluid far exceed those required for the maximal opening of these channels. This situation would result in the excessive influx of Ca^2+^ and consequent cytotoxicity. This issue may be resolved by peripheral cells producing NMDARs that are less sensitive to such stimuli, either through alternate splicing of the mRNA(s) encoding their subunits, distinctive assembles of subunits, or by the presence of additional inhibitory modulators that do not operate in the nervous system [162].

NMDARS are modulated by oxysterols. The most abundant cholesterol metabolite in the brain is 24(S)-HC, which is sometimes referred to as cerebrosterol. This oxysterol potentiates NMDAR-dependent excitatory post-synaptic currents (EPSCs) in rat hippocampal neurons at submicrolar concentrations; an effect that was not caused by unmodified cholesterol, or by other oxysterols, at concentrations up to 10 µM. 24(S)-HC did not modify the gating of other ionotropic glutamate receptors, the GABA receptors and AMPA receptors [164]. SGE-201, a synthetic analogue of 24(S)-HC, influenced the electrophysiological properties of NMDARs present either in single neurons, heterologously expressed in *Xenopus* oocytes, or within hippocampal slices [165]. 24(S)-HC is produced in the central nervous system by the action of *CYP46A1* on cholesterol. In hippocampal slices from *CYP46A1−/−* knockout mice, both the concentration of 24(S)-HC and the basal activity of NMDARs are greatly reduced relative to those from wildtype animals [166]. Use of chimeric receptors to map the binding of positive allosteric modulators to NMDARs indicates that 24(S)-HC interacts with the transmembrane domains of these proteins, at a binding site distinct from those of pregnenolone sulfate, or of docosahexaenoic acid [167]. In mouse hippocampal slices, the GluN2B-selective antagonist Ro25-6981 blocked the 24(S)-HC enhancement of NMDAR-dependent currents, whereas the GluN2A antagonist PEAQX did not [168]. Although 25-HC weakly potentiated NMDAR currents when administered alone, it non-competitively inhibited the potentiation elicited by 24(S)-HC. This suggests the presence of two distinct binding sites for oxysterol modulators within NMDAR complexes [169]. In an oxygen-glucose deprivation model of ischemia, 24(S)-HC exacerbates neuronal death, whereas 25-HC opposes it and can reverse the effects of 24(S)-HC [169]. Another oxysterol congener, 27-HC, reduces NMDAR protein levels in mouse brain, possibly through the downregulation of the activity-regulated cytoskeleton-associated protein (Arc), required for trafficking of these channels to the cell-surface [170].

Negative modulation of NMDARs by some oxysterol congeners might provide an explanation for an absence of cytotoxicity in immune cells, in the face of high concentrations of excitatory amino acids encountered in the blood. NMDARs are present in some immune cells, as reviewed by Boldyrev in 2005 [171]. Using fluorescence-activated cytometry, GluN1 was detected in human peripheral lymphocytes, with the activation of NMDARs in these cells being associated with Ca^2+^ influx, ROS production and cell death. In NK cells derived from this population, NMDA had no effect on IFN-γ production, but suppressed that stimulated by IL-2 [172]. In mouse splenic B-lymphocytes, non-competitive NMDAR antagonists reduced proliferation, chemotaxis and immunoglobulin secretion, but enhanced IL-10 production. However, these antagonists block K^+^ currents through voltage-gated K_v_1.3 and K_Ca_3.1 present in these and other immune cells (see Section 3.2.1.1 and Section 3.3.1) [173,174]. These observations highlight the need for caution when interpreting the effects of small molecule drugs on immune cells.

#### 3.1.3. Ionotropic GABA_A_Rs

As described in Section 2.4, GABA is the predominant central inhibitory neurotransmitter, which exerts its effects via both metabotropic (GABA_B_R) and ionotropic (GABA_A_R) receptors. GABA_A_R complexes are pentameric in structure, with 19 distinct genes encoding subunits, and possess an intrinsic chloride channel [175]. Although no evidence for the effects of oxysterols on the gating of GABA_A_Rs has been reported to date, these channels are modulated by neurosteroids, such as progesterone, allopregnanolone, and their synthetic analogues [176]. GABA_A_R regulates multiple aspects of immune function, including the migration of DC and the activation of T lymphocytes [177].

### 3.2. Voltage-Gated Ion Channels

Voltage-gated ion channels respond to changes in membrane potential in the order of a few tens of millivolts. This allows for the flow of ions, altering membrane potential, in addition to the transfer of electrolytes and, in some cases, the influx of the second messenger Ca^2+^ [15]. This class of channel are often extremely selective in terms of the ions that they conduct, relative to other channel families. Voltage-gated ion channels regulate diverse processes, including secretion, cell motility and the generation of action potentials. Two types of voltage-gated ion channels are known to be modulated by oxysterols: voltage-gated potassium channels and voltage-gated sodium channels.

### 3.2.1. Voltage-Gated Potassium Channels

There is considerable evidence to support roles of voltage-gated K^+^ channels in immune cell function. These channels are present in most cell types and regulate multiple aspects of physiology, including membrane potential, secretion, cell volume and death. They are formed from tetrameric complexes that interact with a range of accessory proteins. Voltage-gated K^+^ channels are diverse in terms of their structures, electrophysiology and pharmacology. To date, of the voltage-gated K^+^ channel family, only K_v_3.1 (belonging to the Shaw, or C, subfamily and encoded by *KCNC1*) and K_Ca_1.1 (Slo-1, or large conductance calcium-activated potassium channel, subfamily M, alpha member 1, encoded by *KCNMA1*) have been reported to be modulated by oxysterols. Since it is gated by a combination of membrane depolarization and increased cytoplasmic Ca^2+^, K_Ca_1.1 will be discussed in greater detail in Section 3.3.1, on channels opened by multiple stimuli.

#### 3.2.1.1. K_v_3.1

The K_v_3 family of voltage-gated K^+^ channels was originally identified as the Shaw family in the fruit-fly *Drosophila melanogaster* and consists of three distinct genes encoding K_v_3.1, K_v_3.2 and K_v_3.3. Electrophysiologically, this family differs from other voltage-gated K^+^ channels in terms of activation at more positive membrane potentials and rapid inactivation. In neurons, these properties mean that K_v_3 channels contribute to rapid, repetitive firing of action potentials and, when mutated, to epilepsy [178]. K_v_3.1 is produced as two alternately spliced variants, K_v_3.1a and K_v_3.1b, differing at the C-terminus. Of these, K_v_3.1b is known to be modulated by oxysterols. In both 158N oligodendrocyte and BV-2 microglial cell lines, incubation with either 24(S)-HC or 7KC for 48 h decreased K_v_1.3b protein abundance and function [179]. K_v_1.3 is responsible for the “I-K^+^ current” and is transcribed at very low levels in cytotoxic T cells, but at high levels in a CD4^-^CD8^-^Thy1^+^ T lymphocytes from mice with autoimmune disease [180].

#### 3.2.2. Voltage-Gated Sodium Channels

Voltage-gated Na^+^ channels have a canonical role in underlying action potentials in excitable tissues, but regulate cell-volume, motility, secretion and death in other cell types. They are comprised of a channel-forming, voltage-sensing α subunit (Na_v_1.1–1.5 encoded by *SCN1A–SCN5A*, Na_v_1.6 encoded by *SCN8A*, Na_v_1.7–1.8 encoded by SCN9A–SCN11A, and Na_x_ encoded by SCN7A) in combination with accessory proteins, such as the β-sub-units (Na_v_β1–4 encoded by *SCN1B–SCN4B*), which influence channel gating or trafficking [181]. Canonical antagonists of these channels include the puffer-fish toxin, tetrodotoxin (TTX), and the small molecule drug, amiloride. Since voltage-gated Na^+^ channels are present in lymphocyte- and monocyte-derived immune cells, there is potential for small molecule modulators of these channels to be utilized as immunotherapeutic drugs. This is offset by potentially severe side-effects of such drugs on the nervous system [182].

There is strong evidence for the modulation of voltage-gated Na^+^ channels by oxysterols [183]. In dorsal root ganglion neurons, the TTX-resistant Na_v_1.8 protein is distributed in clusters, thought to be lipid rafts. This localization was disrupted by either the cholesterol-chelator methyl-β-cyclodextrin, or by the oxysterol 7KC, and was associated with a loss of neuronal excitability [184]. At micromolar concentrations, the oxysterol, cholestane-3β,5α,6β-triol (triol) reduces Na^+^ current density in rat hippocampal pyramidal neurons and ablated seizures in a kainate-induced mouse model of epilepsy [185]. Molecular modelling of voltage-gated Na^+^ channels from the bacteria *Arcobacter butzleri* and *Magnetococcus marinus* indicates that triol binds to the indole ring of tryptophan 122 within the channel pore [186].

As assessed by flow cytometry, the voltage-gated Na^+^ channel inhibitors batrachotoxin or veratridine depolarized a subset of human NK cells [187]. In human HLA-DR-restricted CD4^+^ αβ T cells, stimulation with a non-self-antigenic peptide elicited a Na^+^ current that could be blocked with amiloride, though not with TTX [188]. In contrast, voltage-gated Na^+^ currents detected in human peripheral blood mononuclear cells and in acute lymphocytic leukemia cell lines were largely inhibited by TTX and probably involved Na_v_1.3, Na_v_1.6, or Na_v_1.7. TTX also inhibited the invasiveness of Jurkat and MOLT-4 cell lines [189]. The expression levels of Na_v_1.5 underlie differences in TCR-mediated responses to *Listeria* antigens in two CD4^+^ T-cell lines [190]. Immature dendritic cells (DC) possess voltage-gated Na^+^ channels, mediated predominantly by the Na_v_1.7 subtype, which is downregulated upon differentiation [191].

Voltage-gated Na^+^ channels, mainly those containing Na_v_1.1, Na_v_1.5 or Na_v_1.6, also regulate the physiology of monocytes, macrophages and microglia [192]. Na_v_1.5, a subtype originally identified in the cell-surface membrane of cardiomyocytes, is located in the late endosomes of human monocytes and macrophages, where it permits Na^+^ efflux from these organelles and decreases their pH [193]. In macrophages, these endosomal Na_v_1.5 channels also regulate phagocytosis of mycobacteria [194] and initiation of innate immune responses triggered by double-stranded RNA [195]. In macrophages, alternately-spliced variants of *SCN5A* and *SCN10A* promote the expression and protein abundance of serine/threonine-protein phosphatase 1 regulatory subunit 10 (PPP1R10), a DNA repair enzyme postulated to limit damage to host cells during inflammatory responses [196]. Knockdown of voltage-gated Na^+^ channels using RNA interference reduced proliferation, migration and phagocytosis by the RAW264.7 macrophage cell line. This also reduced macrophage proliferation in atherosclerotic plaques [197].

### 3.3. Ion Channels Gated by Multiple Stimuli

Certain families of ion channels are gated by multiple stimuli. Ion channels known to be modulated by oxysterols are Ca^2+^ and voltage-dependent K^+^ channels and transient receptor potential, canonical, subtype 1 (*TRPC1*) cation channels.

#### 3.3.1. Ca^2+^ and Voltage-Dependent K^+^ Channels

The opening of a subset of voltage-gated K^+^ channels is stimulated by increased cytoplasmic Ca^2+^ concentrations. A well-characterized member of this family is called K_Ca_1.1, or Slo-1, the large-conductance K^+^ channel BK, or MaxiK and is encoded by the calcium-activated K^+^ channel subunit, subfamily M, alpha-1 (*KCNMA1*) gene. BK channels are formed from tetrameric assemblies of K_Ca_1.1 protein and their electrophysiology, trafficking and interactions with ligands are modulated by associations with accessory proteins, most notably β and γ subunits [198]. A key component of K_Ca_1.1 gating is a switch from positively charged amino acids (the “RKK ring”) binding to negatively charged glutamate residues in the protein when in a closed state, to binding oxygen atoms of membrane lipids when in an open state. This gating mechanism helps explain the modulation of BK channel gating by lipophilic modulators [199]. BK gating can be modulated by cholesterol and by bile acids that interact with the channel complex at distinct sites, including at the β1 subunits [200].

Oxysterols modulate BK channel gating. At micromolar concentrations, the side-chain oxysterols 20(S)-HC, 22(R)-HC, 24(S)-HC, 25-HC and 27-HC all inhibited K^+^ currents through K_Ca_1.1, whereas the ring-modified oxysterols, 7-KC or 7α-HC did not [201]. The inhibition of BK by 24(R)-HC is distinct from that exerted by its enantiomer 24(S)-HC in terms of activation kinetics and voltage dependency. The auxiliary subunits of BK channels can also be modulated by oxysterols. Within atherosclerotic lesions, 7KC downregulates KCNB1 (the β1 subunit of BK) [202].

BK channels play important roles in the innate immune system. Monocyte adhesion to endothelial cells elicited by lysophosphatidylcholine is dependent on these K^+^ channels [203]. In macrophages, the blockade of BK inhibits the IκB- and NF-κB-dependent inflammatory pathways stimulated by LPS [204]. BK is not exclusively located in the cell-surface membrane. In macrophages, lysosomal BK channels promote large-particle phagocytosis, by facilitating the release of Ca^2+^ from endolysosomal stores [205]. In the RAW264.7 macrophage cell line, nuclear BK regulates the phosphorylation of the transcription factor cyclic AMP response element binding protein (CREBP) [206].

#### 3.3.2. Transient Receptor Potential Channels

Transient receptor potential channels (TRPs) were originally identified in *Drosophila melanogaster* photoreceptors. In mammals, homologues of this *trp* channel form a family of over thirty genes. They are all homo- or hetero-tetrameric in structure, conducting monovalent cations and in most cases, Ca^2+^. In humans, there are six functional members of the canonical TRP family, *TRPC1-TRPC7*, with *TRPC2* considered to be a pseudogene. Similar to other members of the TRP family, TRPCs are gated by multiple stimuli, often acting in concert. A canonical stimulus for TRP gating is activation of PLC, but neither of the products of this enzyme (IP_3_ and DAG) gate these channels. Depletion of ER Ca^2+^ stores via IP_3_R channels activates TRP channels, possibly by interaction with another class of store-operated Ca^2+^-entry channels, Orai. TRPC1 is also reported to be gated by mechanical stimuli and, when in heterotetrameric complexes with TRPC5, by sphingosine-1-phosphate [207].

There is limited evidence of roles for oxysterols in the regulation of TRPC1 gating. In the THP-1 monocytic cell line 7-KC stimulates translocation of TRPC1 to lipid rafts, with subsequent Ca^2+^-dependent inhibition of the Bcl-2 antagonist of cell death [208]. Similarly, in vascular smooth muscle cells, oxidized low-density lipoprotein (oxLDL) causes cell-death that can be inhibited by TRPC1 knockdown and which is associated with Ca^2+^ influx. Treatment with oxLDL causes an increase in 7KC within the caveolae of these cells, along with the translocation of TRPC1 to these membrane invaginations [209].

TRPC1 has been detected in diverse immune cells. For example, using reverse transcription PCR, in human peripheral blood leukocytes, CD3^+^ T cells and the Jurkat T lymphoma were found [210]. Knockout of TRPC1 in neutrophils impairs their migration and chemotaxis [211]. M1 macrophage polarization is dependent on TRPC1-mediated Ca^2+^ entry [212]. In addition, the inhibition of TRPC1-mediated Ca^2+^ entry promotes ER stress in the RAW264.7 macrophage cell line, initiating inflammatory responses [213]. In primary microglia, soluble factors from the helminth *Mesocestoides corti* inhibit Ca^2+^ influx via TRPC1, suppressing immune responses; oxysterols are candidates for these soluble molecules [214].

## 4. Conclusions and Perspectives

Cells of the immune system contain multiple GPCRs and ion channels that are modulated by oxysterols. However, in many cases, it is unclear if these signalling proteins are influenced by cholesterol oxidation products in vivo under physiological or pathological settings. Similarly, many of the consequences of oxysterol modulation of GPCRs and ion channels within the immune system have not been investigated. Furthermore, the molecular mechanisms by which oxysterols alter the function of many of these receptors awaits full characterization. Consequently, there is considerable scope for exploration of cell-surface receptors for oxysterols in the immune system, both for scientific discovery and for the development of new immunotherapeutic regimes.

## Figures and Tables

**Figure 1 cells-10-02078-f001:**
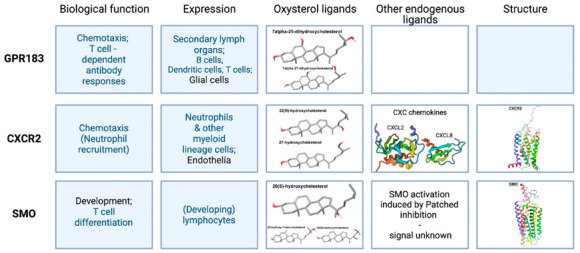
Overview of the biological functions and distribution (Expression)s of oxysterol activated GPCRs in cells of the immune system. The structures of their endogenous oxysterol ligands, details of other endogenous ligands and, where available, models of their three-dimensional structures of these receptors are also shown. This image was created using BioRender (BioRender.com).

**Figure 2 cells-10-02078-f002:**
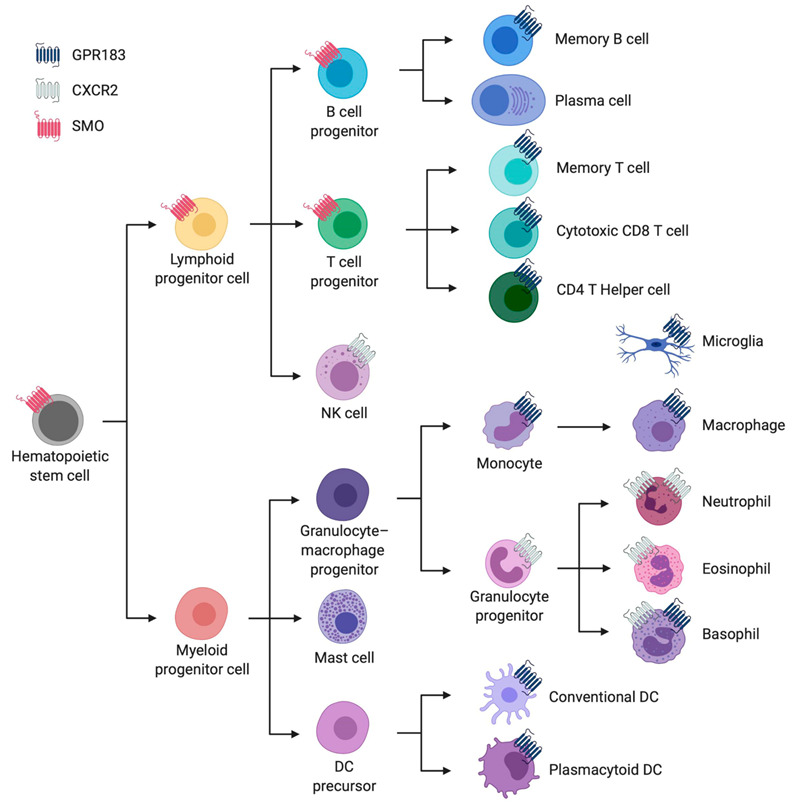
Overview of the main immune cell types from the hematopoietic lineage with an abstraction of their cellular expression of the three GPCRs discussed in this article. These GPCRs are represented by three distinct colours: purple for GPR183, light grey for *CXCR2* and red for *SMO*. This overview was created in BioRender (BioRender.com) and is based on a consensus from four independent transcriptomic data sets from human immune cells [37,38,39], as summarized in Appendix A.

**Figure 3 cells-10-02078-f003:**
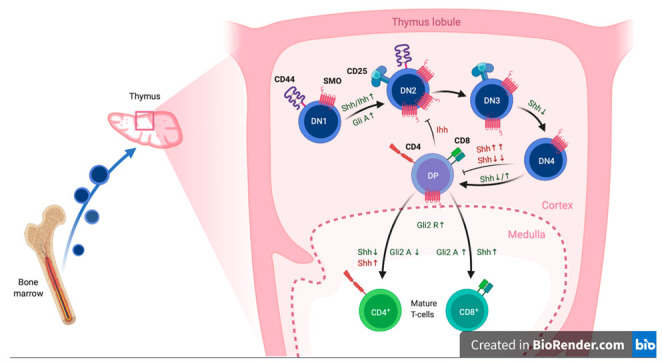
Summary of the differential expression and signalling effects of SMO, including its upstream signals (Hh) and its downstream transcription factors (Gli) on T-cell development in the thymus. Inhibitory signals and effects are highlighted in red, differentiation stimulating effects are marked green. This figure was adapted from “T-Cell Development in Thymus 2”, by BioRender.com (2021), retrieved from https://app.biorender.com/biorender-templates.

**Figure 4 cells-10-02078-f004:**
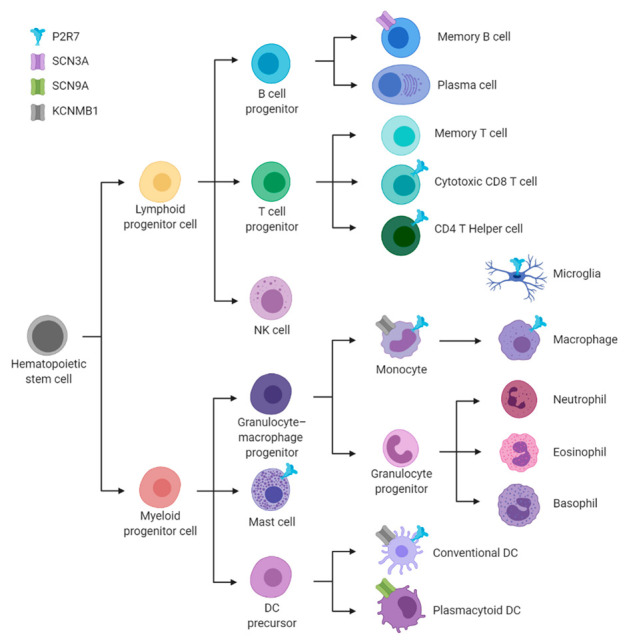
Overview of the key immune cell types from the hematopoietic lineage with an abstraction of their cellular expression of the ion channels discussed in this article. These channels are represented by icons of different colours: blue Figure 2. X7R, purple for SCN3A, green for SCN9A and grey for KCNMB1. This overview is derived from consensus of four independent transcriptomic data sets from human immune cells [31,32,33], as summarized in Appendix A. Figure 4 was created in BioRender (BioRender.com).

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
