# Peer review of "Multiple Targets for Oxysterols in Their Regulation of the Immune System"

_cells, 2021, doi:10.3390/cells10082078_

Round 1

Reviewer 1 Report

In this manuscript, the authors reviewed the two broad classes of cell surface receptors for oxysterols, G-protein coupled receptors and ion channels.  The authors believe that the mechanism by which oxysterols play an important role in physiology and pathophysiology.  This review also highlighted that oxysterols have potential for physiological and therapeutic modulation of the immune system.  Overall, the review is comprehensive and helpful for readers in the oxysterol field.  However, the manuscript has missed key information, which has been recently published in iScience, Metabolism Clinical and Experimental, and J.L.R.  The global regulatory functions cannot be interpretated by activation of nuclear receptors.  These recent publications have shown that oxysterols serve as endogenous epigenetic regulators, activating DNA methyl transferase, by which oxysterols up- and down-regulate CaVs-AMPK and MAPK signaling pathway.  Furthermore, oxysterols can be sulfated and the sulfated oxysterols inhibit DNA methyl transferases. Therefore, oxysterols with oxysterol suflates co- ordinately maintenance inflammatory responses and play an important role in many physiological and pathophysiological events.  More importantly, oxysterol sulfates have been used in clinical trials.  This manuscript needs to cite and discuss these discoveries for updated information. 

Author Response

Response to Reviewer 1

Reviewer 1 comments: “In this manuscript, the authors reviewed the two broad classes of cell surface receptors for oxysterols, G-protein coupled receptors and ion channels.  The authors believe that the mechanism by which oxysterols play an important role in physiology and pathophysiology.  This review also highlighted that oxysterols have potential for physiological and therapeutic modulation of the immune system.  Overall, the review is comprehensive and helpful for readers in the oxysterol field.  However, the manuscript has missed key information, which has been recently published in iScience, Metabolism Clinical and Experimental, and J.L.R.  The global regulatory functions cannot be interpretated by activation of nuclear receptors.  These recent publications have shown that oxysterols serve as endogenous epigenetic regulators, activating DNA methyl transferase, by which oxysterols up- and down-regulate CaVs-AMPK and MAPK signaling pathway.  Furthermore, oxysterols can be sulfated and the sulfated oxysterols inhibit DNA methyl transferases. Therefore, oxysterols with oxysterol suflates co- ordinately maintenance inflammatory responses and play an important role in many physiological and pathophysiological events.  More importantly, oxysterol sulfates have been used in clinical trials.  This manuscript needs to cite and discuss these discoveries for updated information.”

We thank the reviewer for their valuable suggestions and have modified the revised manuscript to include a new paragraph describing the epigenetic actions of 25-HC and 25-HC3S.

Reviewer 2 Report

The present review attempts to provide a summary of published literature that describe cell surface receptors for oxysterols that regulate cells of the immune system.   The review is extremely hard to read due to lack of flow and logical organization.  It is best if the authors could ask themselves what is the main purpose of this review and then put efforts into reorganizing the content in order to make better sense to the readers.  The following are a few specific comments and suggestions that hopefully will help the authors:

  1. The current title is not appropriate and it should specify the main focus of the review on cells of the immune system.
  2. In the Introduction section of the manuscript the use of the term "secondary oxysterols" is not appropriate and is confusing (it sounds like the oxysterols are secondary metabolites).  The authors should consider simply stating that oxidation of cholesterol can occur on the sterol rings, the side chain, or both.  
  3. Figure 1 is confusing and it appears to show that 20S-hydroxycholesterol is an inhibitor of Ptch, which is not correct. And what do they mean by "canonical ligands"?  Do they mean endogenous ligands?  All figures can benefit from a more clear and thoroughly described figure legend in order to explain the diagrams.
  4. Figure 2 is also confusing.  The authors appear to want to show that monocytes do not express Smo on their cell surface but gain such expression after differentiating into macrophages.  If this has been shown in previously published reports, such reports should be clearly identified.  
  5. This reviewer is not convinced that providing the route of synthesis of potential oxysterol ligands for the cell surface receptors described is helpful or useful.  If anything it makes the review unnecessarily cluttered and hard to follow.  

Author Response

Responses to Reviewer 2

Reviewer Comments: “The present review attempts to provide a summary of published literature that describe cell surface receptors for oxysterols that regulate cells of the immune system. The review is extremely hard to read due to lack of flow and logical organization. It is best if the authors could ask themselves what is the main purpose of this review and then put efforts into reorganizing the content in order to make better sense to the readers. The following are a few specific comments and suggestions that hopefully will help the authors:”

We thank the reviewer for their insights and agree that their suggestions will improve our review. We have modified the manuscript accordingly.

  1. “The current title is not appropriate and it should specify the main focus of the review on cells of the immune system”.

We have revised the title accordingly.

  1. “In the Introduction section of the manuscript the use of the term "secondary oxysterols" is not appropriate and is confusing (it sounds like the oxysterols are secondary metabolites). The authors should consider simply stating that oxidation of cholesterol can occur on the sterol rings, the side chain, or both.”

We agree with the reviewer and have modified this sentence to improve its clarity.

  1. “Figure 1 is confusing and it appears to show that 20S-hydroxycholesterol is an inhibitor of Ptch, which is not correct. And what do they mean by "canonical ligands"? Do they mean endogenous ligands? All figures can benefit from a more clear and thoroughly described figure legend in order to explain the diagrams.”

The figures and their accompanying legends have been revised, as suggested by the reviewer.

  1. “Figure 2 is also confusing. The authors appear to want to show that monocytes do not express Smo on their cell surface but gain such expression after differentiating into macrophages. If this has been shown in previously published reports, such reports should be clearly identified.”

We agree with the reviewer that the four independent, published transcriptomic data-sets on immune cells on which Figure 2 is based do not support expression of SMO in macrophages, in contrast to other reports in the literature. We therefore analysed the transcriptional programs of macrophages activated with 28 different stimuli including pattern recognition receptor ligands, cytokines and metabolic cues published by Xue et al (Immunity 2014, PMID: 24530056). This very comprehensive study did not detect expression of SMO in macrophages and we therefore modified Figure 2 by removing SMO from macrophages.

“This reviewer is not convinced that providing the route of synthesis of potential oxysterol ligands for the cell surface receptors described is helpful or useful. If anything it makes the review unnecessarily cluttered and hard to follow.”

Although we agree with the reviewer in terms of the complexity of the synthesis of some endogenous oxysterols, there is still value in retaining most of these details. This is particularly pertinent in terms of understanding the role of GPR183 in lymphocyte migration and differentiation. Consequently, we have made only minor modifications to the text related to this aspect.

Reviewer 3 Report

The manuscript by Reinmuth et al is clear, well written and I would suggest only to improve resolution of supplementary figures.

Author Response

Response to Reviewer 3

Reviewer Comments: “The manuscript by Reinmuth et al is clear, well written and I would suggest only to improve resolution of supplementary figures.”

We thank the reviewer for their positive assessment. The resolution of the supplementary figures has been increased.

Round 2

Reviewer 2 Report

The authors have adequately responded to this reviewer's comments and the manuscript is not acceptable for publication.